# Rewilding the Detroit, Michigan, USA–Windsor, Ontario, Canada Metropolitan Area

**John H. Hartig**

Great Lakes Institute for Environmental Research, University of Windsor, Windsor, ON N9B 3P4, Canada; jhhartig@uwindsor.ca

**Abstract:** Rewilding attempts to increase biodiversity and restore natural ecosystem processes by reducing human influence. Today, there is growing interest in rewilding urban areas. Rewilding of the Detroit, Michigan, USA and Windsor, Ontario, Canada metropolitan area, and its shared natural resource called the Detroit River, has been delineated through the reintroduction of peregrine falcons and osprey, and a return of other sentinel species like bald eagles, lake sturgeon, lake whitefish, walleye, beaver, and river otter. Rewilding has helped showcase the value and benefits of environmental protection and restoration, ecosystem services, habitat rehabilitation and enhancement, and conservation, including social and economic benefits. Improved ecosystem health and rewilding have become a catalyst for re-establishing a reconnection between urban denizens and natural resources through greenways and water trails. The provision of compelling outdoor experiences in nature, in turn, can help foster a personal attachment to the particular place people call home that can help inspire a stewardship ethic.

**Keywords:** ecological restoration; urban areas; natural resources; conservation



## 1. Introduction

Rewilding emerged as a conceptual framework in North America in the 1980s, when it was originally called "wilderness recovery" [1]. Rewilding attempts to increase biodiversity and restore natural ecosystem processes by reducing human influence [2]. Considerable interest in rewilding exists throughout the world [3–6]. One of the most well-known examples of rewilding in North America is the reintroduction of grey wolves (*Canis lupus*) to Yellowstone National Park, Wyoming, USA.

In Yellowstone, wolves had become villainized as a danger to humans and a menace to ranchers, and wolf predation on elk (*Cervus canadensis*) was viewed as "wanton destruction" of more desirable species [7]. Their critical role in achieving an ancestral Yellowstone ecosystem was not well-understood. Predator controls, including the poisoning of wolves, began in the park in the late 1800s. Over time, more than 130 wolves were killed under the guise of conservation, with the last report of a wolf killing occurring in 1926 [8]. The elimination of this top predator disrupted the historical food web, allowing elk to increase in abundance which increased elk herbivory resulting in a decline of plant species like aspens, willows, and grasses [9].

Then in 1995, 14 Canadian wolves were reintroduced into the park, helping restore a more functional ecosystem [10,11]. Fifteen years after wolf re-introduction, monitoring has documented substantial food web changes, including a decreased elk population, increased numbers of beaver and bison, and increased aspens and willows [9].

The concept of rewilding has evolved from its initial emphasis on protecting large habitat tracts and reintroducing top predators to an adaptive management approach that assesses, sets priorities, and takes action in an iterative fashion for continuous improvement in ecosystem health [12]. Today, there is a growing interest in urban rewilding and ecology. Good urban examples of fostering urban biodiversity and rewilding include Singapore

in Southeast Asia [13]; Toronto, Ontario, Canada [14]; and Chicago, Illinois, USA [15]. Urban rewilding often includes reintroducing native plant and/or animal species, building green infrastructure, building greenways with concomitant habitat features, cleaning up industrial brownfields for city parks with native landscaping, or creating urban gardens or incorporating green features into building design that often includes use of non-native species. This paper presents a case study of urban rewilding and ecological restoration initiatives in the Detroit, Michigan, USA–Windsor, Ontario, Canada metropolitan area, based on long-term monitoring of ecosystem health indicators, and summarizes challenges, lessons learned, and benefits.

## 2. Study Area and Background

Detroit, Michigan and Windsor, Ontario are adjacent border cities on the Detroit River that flows into western Lake Erie (Figure 1). Their strategic location on the Detroit River, with access to the Great Lakes, helped enable the growth and development of Detroit and Windsor into the "automobile capitals" of the United States and Canada, respectively. The current populations of the Detroit and Windsor metropolitan areas are 4,800,000 and 342,000, respectively [16].

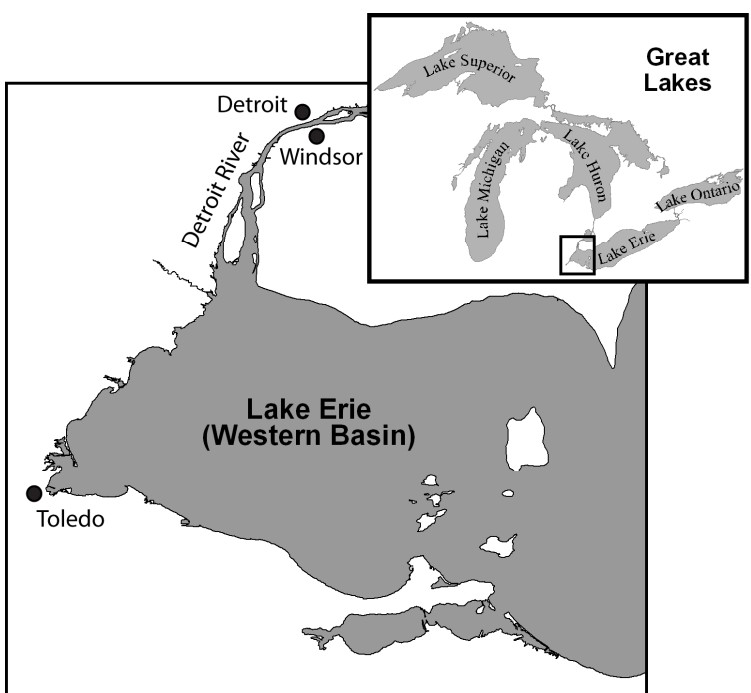

**Figure 1.** Detroit, Michigan and Windsor, Ontario are border cities on the Detroit River (42.184863°; −83.057444°) that connect the upper and lower Great Lakes.

Both Detroit and Windsor have faced numerous urban challenges, including urban sprawl, land use changes, the downturn of industry resulting in being portrayed as "rust belt" cities, and economic, social, and environmental crises. The response to the environmental crises would eventually lay the foundation for urban rewilding.

The Detroit River is a 51 km strait or connecting river system through which water from North America's upper Great Lakes—Lakes Superior, Michigan, and Huron—flows into the lower Great Lakes—Lakes Erie and Ontario. Ninety-five percent of the flow of the Detroit River comes from the upper Great Lakes and 80% of the inflow to Lake Erie comes from the Detroit River [17]. During the 1960s, the Federal Water Pollution Control Administration (predecessor of the U.S. Environmental Protection Agency) identified the Detroit River as one of the most polluted aquatic ecosystems in the United States [18]. Examples of water pollution and natural resource degradation include:

- During the 1960s, Detroit's wastewater treatment plant was only achieving primary treatment (i.e., removal of material that would float or settle out and disinfection), and its regional storm and sanitary sewer system was discharging approximately 117.3 billion liters of untreated wastewater per year from combined sewer overflows;
- During the 1960s, oil pollution of the Detroit River was substantial, causing the death of 12,000 and 5400 waterfowl in 1960 and 1967, respectively;
- The Rouge River (a tributary of the Detroit River) caught on fire in 1969 as a result of oil pollution;
- The fishery had to be closed in 1970 due to mercury contamination (i.e., the Mercury Crisis of 1970);
- The International Joint Commission designated Detroit and Rouge Rivers as pollution hotspots called Great Lakes Areas of Concern and called for the development and implementation of cleanup plans to restore impaired beneficial uses; and
- Severe algal blooms occurred in western Lake Erie during the 1950s–1980s (they later diminished in the late 1980s and early 1990s, and returned again in the late 1990s–2020s [16].

### 3. Long-Term Monitoring of Key Indicators of Ecosystem Health

The State of the Strait is a Canada–U.S. biennial forum that alternates between the two countries and brings together resource managers, academic and governmental researchers, business leaders, representatives of environmental and conservation organizations, students, and concerned citizens to foster and advance ecosystem-based management of the Detroit River and western Lake Erie (shared natural resources of the two countries). The conference now has a more than 25-year history [19].

The Detroit River–Western Lake Erie Indicator Project was initiated in 2018 to compile long-term data sets (most with 40 or more years) on 61 indicators of ecosystem health and co-produce knowledge to strengthen science-policy-management linkages [16]. Discrete indicator reports were prepared by individuals from research institutions, management agencies, or nongovernmental organizations who collected and managed the data sets. These indicator reports were then presented at the 2019 State of the Strait Conference for discussion and peer review.

### 4. Environmental Cleanup Catalyzes Rewilding

Public outcry over pollution in the 1960s led to the establishment of the 1970 Canada Water Act, the 1970 U.S. National Environmental Policy Act, the 1972 U.S. Clean Water Act, the 1972 Canada–U.S. Gret Lakes Water Quality Agreement, the 1973 U.S. Endangered Species Act, and many complementary state and provincial environmental laws. Long-term monitoring programs have documented substantial environmental improvements in the Detroit River since the passage of these laws and signing of the Agreement nearly 50 years ago, including reductions in critical pollutant loadings, upgraded municipal wastewater treatment from primary to secondary treatment with phosphorus removal, reduced contaminant levels in fish and wildlife, remediation of some contaminated sediment hot spots, and enhancement or rehabilitation of targeted habitats [16]. These environmental improvements have resulted in ecological recovery and rewilding. It should also be noted that declining human population in Detroit and increasing vacant land have contributed to rewilding. Presented below is evidence of rewilding of this major urban-industrial area.

### 5. Avian Species

Good avian examples of rewilding in this major metropolitan area are the reintroduction of peregrine falcons (*Falco peregrinus*) and osprey (*Pandion haliaetus*) (Table 1). By the 1960s, peregrine falcons had experienced a population crash in the Detroit River watershed and no young were being produced because of egg-shell thinning caused by organochlorine pesticides like DDT [20]. The State of Michigan responded by banning the pesticide DDT in 1969, followed by a nationwide ban in 1972.

In response to this peregrine falcon population crash, they were placed on the federal "endangered" species list in 1970. The Michigan Department of Natural Resources then initiated a recovery program that released five young in hacking boxes—wooden boxes designed to simulate nests—on downtown Detroit skyscrapers in 1987 [20]. No young were produced in the first five years; however, two young were produced in 1993, followed by a slow increase in productivity until 30 young were fledged in 2016 [20]. The peregrine falcon was removed from the "endangered" species list in 1999.

Similar to peregrine falcons, the osprey population in Michigan experienced a dramatic population decline as a result of organochlorine pesticide use [21]. By 2002, the Michigan Department of Natural Resources reported that there was only one active nest in southern Michigan [21]. As part of recovery efforts, osprey was first reintroduced in the late 1990s in metropolitan Detroit by the Michigan Department of Natural Resources and Osprey Watch of Southeast Michigan (now called Michigan Osprey). Annual citizen science monitoring has documented 50 or more nesting pairs in 2016 and 2017, confirming that the osprey recovery target of 30 nesting pair in the southern Lower Peninsula has been achieved [21]. In response, the Michigan Department of Natural Resources upgraded osprey from "threatened" to "Species of Greatest Conservation Need". Federally, it is also protected under the Migratory Bird Treaty Act of 1918.

Similar to peregrine falcons and osprey, the bald eagle (*Haliaeetus leucocephalus*) population in the Detroit River watershed collapsed and no young were produced for 26 years starting in 1961 as a result of organochlorine pesticide use (Table 1). In response, the bald eagle was identified as a "threatened species" in Michigan in 1978 and later elevated to "endangered". As organochlorine pesticide exposure diminished and conservation practices instituted, bald eagles returned in 1987 and by 2015 monitoring documented the presence of 29 active bald eagle nests in this urban area [22]. It was removed from the federal "endangered" species list in 2007. It is also recognized that birds of prey also benefit from prey availability.

Michigan's native wild turkey (*Meleagris gallopavo*) population was decimated by overhunting and habitat loss, and was declared extirpated after the last bird was sighted in Van Buren County in 1897 [23]. The cause of this population collapse was unregulated hunting and habitat loss. They were reintroduced into southwestern Michigan in the 1950s, but it was the collective conservation efforts of hunters and the Michigan Department of Natural Resources, starting in 1983, that brought the population back to an estimated 200,000 wild turkeys inhabiting Michigan in the early 2020s [24]. Today, these birds have adapted well to urban and suburban areas like metropolitan Detroit [23].

**Table 1.** Evidence of rewilding in the watershed of the Detroit River based on long-term ecosystem health indicator trend data [9,12]. Re-introduced species noted with an *.

| Indicator | Evidence of Rewilding | References |
|---|---|---|
| Peregrine falcon * | They were extirpated from Michigan by the mid-1960s and re-introduced in Detroit in 1987. By 2015, 30 young were fledged in metropolitan Detroit. | [20] |
| Osprey * | A population crash occurred in the 1960s, reaching a low point in 2002 when only one active next was reported in southern Michigan. They were reintroduced into metropolitan Detroit in the 1990s and now are thriving, with 38, 50, and 52 nesting pairs reported in 2015, 2016, and 2017, respectively. | [21] |
| Bald eagle | For 26 years (1961–1987) no bald eagles were fledged in metropolitan Detroit. Then the population rebounded, fledging 28–38 young per year during 2011–2015. | [22] |
| Wild turkey * | They were reported as extirpated in Michigan in 1897. A population recovery program was initiated in 1986 and today they are found throughout Michigan but are noticeable for expanding their range into metropolitan Detroit. | [23] |

**Table 1.** *Cont.*

| Indicator | Evidence of Rewilding | References |
|---|---|---|
| Lake whitefish | Spawning runs into the Detroit River disappeared by 1916. The return of spawning in the river was documented in 2006, after a 90-year absence. | [25] |
| Lake sturgeon | Based on fishery monitoring and assessments, no lake sturgeon reproduction was reported in the Detroit River for nearly three decades (1970s–1999). After a nearly 30-year absence, lake sturgeon spawning has again been documented in the river and the population is now estimated to be over 4400 individuals. | [26] |
| Walleye | The Lake Erie Committee of the Great Lakes Fishery Commission estimated the Lake Erie population to be approximately 10 million age-2+ walleye in 1978. Forty years later (2018), the population was estimated to be approximately 40 million age-2+ walleye. | [27] |
| Beaver | They were last reported in the Detroit River in 1877, after which they were declared extirpated. They returned in 2008 after a 130-year absence and have now been observed in at least six watershed locations. | [28] |
| River otter | They were extirpated from the Detroit River by the early 1900s. Following reintroduction in eastern Lake Erie tributaries in Ohio in 1986, they slowly expanded their range and were documented in the Detroit River in 2022, representing the first time in more than 100 years. | [29] |
| Coyote | They are not native to Michigan. The first report of a coyote in Michigan was from Washtenaw County in the 1890s. Over time, they have gradually expanded their range, especially in urban areas, and are now frequently sighted in the metropolitan Detroit area. | [23] |
| Wildcelery | Between 1950–1951 and 1984–1985, widcelery tuber density in the Detroit River decreased by 72%. Then, between 1984–1985 and 1996–1997, tuber density increased by 251%. | [30] |
| Tree cover | Both Essex County, Ontario and southeast Michigan have experienced substantial loss of forest or tree cover. Although Essex County forest cover has increased from a historical low of less than 4% to 5.7% in 2023, it is significantly below the target of 15%. Tree canopy area in southeast Michigan is currently at 33% and below the target of 40%. | [31,32] |
| Invasive species | Invasive species are a problem on both sides of the river. Although removal of invasive species is not a form of rewilding, it does result in the return of native species in urban areas. Southeast Michigan's nonprofit organization called The Stewardship Network engages people in citizen science, control of invasive species, and rehabilitation of habitats. Throughout 14 years of its Spring Challenge, more than 39,000 people have been involved in removing 1.09 million kg of invasive species, equaling USD 11 million in work hours. Similar stewardship activities are underway in Essex County, Ontario under the direction of Essex Region Conservation Authority. | [33] |
| Soft shoreline | Urban and industrial waterfront development has resulted in substantial shoreline hardening. On the Canadian side, 61% of the shoreline has been surveyed and found to be soft, with a target of at least 70% to achieve good quality, and 13 soft shoreline projects were completed since the late 1990s. On the U.S. side, 43% of the shoreline has been found to be soft, with the same 70% target, and 39 soft shoreline projects were completed since 2000. | [34,35] |

## 6. Fishes

Lake whitefish (*Coregonus clupeaformis*), lake sturgeon (*Acipenser fulvescens*), and walleye (*Sander vitreus*) are all lithophilic spawners. Starting in 1874 and continuing through 1916, considerable rock spawning habitat in the Detroit River was lost during the construction of deep-water shipping channels (i.e., the Livingstone and Amherstburg Channels) [36]. The large lake whitefish spawning runs into the Detroit River that characterized the late 1800s and early 1900s disappeared by 1916 (Table 1). Fishery managers determined the cause to be overfishing, predation by and competition with invasive species, degradation of water quality and habitat, and the loss of a shrimplike food source called *Diporeia* [25].

As environmental protection programs started improving the quality of the Detroit River, fishery biologists hypothesized that lithophilic spawning fish productivity was now more limited by habitat than environmental quality. In response, 10 rock spawning reefs have been constructed in the Detroit River since 2003 [37,38]. In 2005, fishery biol-

ogists documented lake whitefish spawning again in the Detroit River for the first time since 1916 [25].

The waters from southern Lake Huron to western Lake Erie were one of the most productive areas in North America for lake sturgeon in the late 1800s. Lake sturgeon then experienced a similar population crash to lake whitefish, due to overfishing, loss of spawning habitat, and water pollution [26].

For nearly three decades, starting in the 1970s, fishery surveys in the Detroit River found no lake sturgeon spawning (Table 1). Lake sturgeon spawning was then documented in 2001 on a coal cinder pile near Detroit River's Zug Island, representing the first time in nearly 30 years [26]. The U.S. Fish and Wildlife Service has estimated the current Detroit River sturgeon population size, based on mark-and-recapture studies, at over 4400 individuals and characterized it as a large and self-sustaining population (Table 1).

The Detroit River, along with Lake Erie, shares the distinction of having some of the best walleye fishing in North America. Massive runs of walleye, as many as 10 million fish, ascend the Detroit River each spring to spawn on its rock substrates, creating a world-class fishery; however, it was not always that way.

The Lake Erie Committee of the Great Lakes Fishery Commission is the primary institutional mechanism for cooperative monitoring, assessment, research, and management of the Lake Erie fishery. In 1978, this committee declared the walleye population of Lake Erie to be in a "crisis state" due to overfishing, habitat degradation, and water pollution [27]. Although there is much year-to-year variability in population estimates, the 2018 population size was four times higher than when it was declared in a "crisis state" in 1978 (Table 1). The local fishing economy supported by these walleye runs is substantial, bringing in more than USD 1 million to local Downriver communities each spring [27].

## 7. Mammals

Beaver (*Castor canadensis*) was extirpated from the Detroit River watershed in 1877 as a result of overharvesting during the fur trade era (i.e., 17th and 18th centuries) and loss of habitat [28]. River otter (*Lontra canadensis*) was extirpated from this watershed in the early 1900s as a result of over-harvesting during the fur trade era and then loss of habitat and pollution from urbanization [29]. It should be noted that during peak oil pollution of the Detroit River in the 1940s–1970s, these two mammals could not have survived because oil would mat their fur and they could not thermoregulate.

In 2008, beaver returned to the Detroit River for the first time in 131 years and can now be seen in numerous locations in the watershed (Table 1). In 2023, river otter returned to the Detroit River for the first time in more than 100 years (Table 1).

Coyotes (*Canis latrans*) are not native to Michigan and were likely introduced in southeast Michigan's Washtenaw County in the 1890s [23]. They are highly adaptable and are known to thrive in a wide range of environments, including urban areas like metropolitan Detroit where they are now commonly seen [22].

## 8. Plants

Wildcelery (*Vallisneria americana*) is a submersed aquatic plant that is a very important food for diving ducks in the Detroit River and is considered an indicator of ecosystem health. Wildcelery tuber density increased in the Detroit River between 1984–1985 and 1996–1997 by 251% [30]. It is believed that this tuber increase is due to increased water clarity in the river resulting from water pollution control and the presence of exotic zebra (*Dreissena polymorpha*) and quagga mussels (*D. bugensis*) that filter particulate matter [30].

Loss of tree cover is a ubiquitous problem in urban areas. There is growing interest in urban reforestation to provide habitat for other species, reduce stormwater runoff during wet-weather events, absorb carbon dioxide, and provide a sound buffer for unwanted noise. Currently, 33% of metropolitan Detroit is covered in tree canopy, with a target of increasing this to 40% (Table 1). Since 1989, the nonprofit organization called Greening of Detroit has planted more than 135,000 trees in Detroit, Hamtramck, and Highland Park.

Over the next five years, American Forests and its partners will be planting 75,000 trees in Detroit. Despite Essex Region Conservation Authority planting over six million trees in Essex County, Ontario since 1976, county forest cover is only 5.7% and well below the target of 15% (Table 1).

The Stewardship Network of Michigan helps connect, equip, and mobilize people and organizations to care for land and water in their communities, with a priority on metropolitan Detroit. In 15 years of this network's Spring Challenge, more than 39,000 people have been engaged in removing 1.09 million kg of invasive species, equaling USD 11 million in workhour value [33]. Similar stewardship activities are underway in Essex County, Ontario under the direction of the Essex Region Conservation Authority.

Hard shorelines are where concrete breakwaters or steel sheet piling are used to achieve shoreline stabilization and safety; however, they provide no habitat. In contrast, soft shorelines utilize rocks, plants, and other materials to achieve stability while enhancing habitats and improving aesthetics [34]. On the Canadian and U.S. sides of the Detroit River, 61% and 43% of the shorelines are currently considered soft, respectively (Table 1). The target is to achieve at least 70% for a healthy ecosystem.

It should be noted that in the Detroit–Windsor metropolitan areas, there are concerted efforts to create green infrastructure (i.e., rain gardens, pollinator gardens, wildflower gardens, pervious parking lots, green roofs, bioswales, etc.) to help address urban stormwater runoff from increasing intensity and frequency of storms caused by climate change and to create habitats for other species as part of "greening" these urban areas.

## 9. Remaining Ecosystem Challenges and Lessons Learned

Although there has been considerable environmental improvement in the Detroit River and evidence of rewilding, much remains to be done to reach long-term ecosystem targets. Key challenges include mitigating and adapting to climate change, addressing stormwater and sewage overflows, preventing pollution, remediating contaminated river sediments and brownfields that are stymying further improvement in ecosystem health, rehabilitating and conserving habitats, and preventing the introduction of invasive species [16,18].

Lessons learned include: a clear and compelling vision for a healthy and biodiverse city must be developed, agreed to, and carried in the hearts and minds of all stakeholders; monitoring is the foundation of sound natural resource management and must continue to be a priority to practice adaptive management; co-production of knowledge and co-innovation of solutions are essential to achieve a healthy and biodiverse city; and natural resource and biodiversity champions need to be at the table where urban redevelopment projects are discussed and developed so that there is a voice for other species and their habitats.

## 10. Benefits of Rewilding and Concluding Thoughts

Benefits of urban rewilding include enhancing ecosystem services, reversing biodiversity loss, sequestering carbon, mitigating extreme weather events, and combating urban sprawl [39]. Social and economic benefits of urban rewilding are equally important, including improving human health and well-being, expanding outdoor recreation and its outdoor recreational economy, furthering urban place-making, and strengthening communities [40].

Historically, Detroit's and Windsor's waterfronts were dominated by industrial and commercial activities, and people lost their physical connection to the river [41]. As environmental protection and pollution control programs started improving ecosystem health of the Detroit River and resulting in rewilding, concerned citizens started calling for improving public access to the river, including establishing linked riverfront greenway trails (i.e., nonmotorized trails that are established for recreational use and environmental and natural resource protection) and water trails for canoeing and kayaking.

In response to increasing public demand for access to the waterfront and non-motorized modes of transportation, greenway systems started first in Windsor in the 1960s and 1970s, and then in Detroit in the 1990s [41]. Today, both cities have waterfront greenways that are connected to additional greenway systems that circumnavigate each city. Incidentally,

Detroit's Riverwalk has been identified as the No. 1 riverwalk in the United States by USA Today three years in a row. Such efforts to reconnect urbanites with land and water through greenways and water trails, including showcasing rewilding, habitat rehabilitation and enhancement, ecosystem services, and conservation, are helping make nature part of daily routines [42,43]. This, in turn, will help develop a personal attachment to the particular place people call home that can help inspire a stewardship ethic [28]. Research has shown that successful cities, by integrating nature with culture, meet the universal need to maintain or restore contact with nature and reap its many benefits [44].

**Funding:** This research was funded by the University of Windsor and the sponsors of the State of the Strait Conference [16].

**Institutional Review Board Statement:** Not applicable.

**Data Availability Statement:** Data used in the study were compiled through the State of the Strait Conference [16] and can also be found here: https://www.uwindsor.ca/glier/419/current-key-indicator-reports (accessed on 17 August 2023).

**Acknowledgments:** This project was made possible by more than 50 organizations that provided long-term data sets on 61 indicators of ecosystem health and the 2022 State of the Strait Conference sponsors [16]. Without their contributions and support, this project would not have been possible.

**Conflicts of Interest:** The author declares no conflict of interest. State of the Strait Conference sponsors had no role in the selection of indicators, monitoring protocols, data collection or interpretation, preparation of the manuscript, or the decision to publish the results.

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
