# Peer review of "Rewilding the Detroit, Michigan, USA–Windsor, Ontario, Canada Metropolitan Area"

_resources, doi:10.3390/resources12100117_

Round 1

Reviewer 1 Report

This is a worthwhile paper on the 'rewilding' of the Detroit/Windsor urban area that provides much opportunities and hopes for a more sustainable future for coexistence of man and nature.

My main problems concern the use of the term 'rewilding', the naive us of the term 'balanced' and the failure to recognize the driving forces behind the described 'rewilding'. 

First, spontaneous recovery and recolonization, shrubification and 'weedification' and expansion of non-native flora/fauna is not what is typically referred to as 'rewilding'. Author need to adjust/moderate those sections accordingly. 

Second, there is nothing like 'balance' in the wild, its all processes. The author need to moderate and/or adjust those statement (see suggestions as comments in attached pdf).

Third, author must include reflections over other processes than 'rewilding' to describe how urban areas has been reconquered, as for example house/neighborhood abandonment, land abandonment, economic crisis, changed hunting practices.

Finally, I think "Soulé ME, Noss RF (1998). Rewilding and biodiversity: complementary goals for continental conservation. Wild Earth. (1998) 8:18–28." should be ackrediterad for the coining of 'rewilding'.

Se additional comments in pdf.

Author Response

Addressed editorial suggestions in paragraphs 2-5.

Section 4 - added sentence noting that declining human population in Detroit and increasing vacant land have contributed to rewilding. 

Section 5, 4th paragraph - Added sentence reflecting that birds of prey also benefit from prey availability.

Section 5, 5th paragraph - Added sentence noting that the cause of the population collapse was both unregulated hunting and habitat loss. 

Table 1 - Noted reintroduced species with an *.

Regarding the note that control of invasive species is not rewilding, I agree. It is, however, an important part of urban rewilding in that it allows for the return of native species. This has now been noted in Table 1.

Added a new paragraph in Section 2 describing some of the many urban challenges facing Detroit and Windsor and how the response to environmental crises would eventually lay the foundation for urban rewilding. 

Accredited Soulé ME, Noss RF (1998).

Reviewer 2 Report

Interesting communication of rewilding project. I would suggest to include some references to other studies, for example in the introduction, such as examples from countries in Europe, Asia or in other parts of the world. 

Author Response

I added four rewilding citations from Asia, Africa, Australia, and the Scottish Highlands in the Introduction. 

Reviewer 3 Report

This is exceptionally good paper. I recommend to publish it with very minor edits. 

General comments

The manuscript entitled “Rewilding the detroit, Michigan, USA-Windsor, Ontario, Canada Metropolitan Area” by J H Hartig is exceptionally good and informative study about the experience of nature restoring in USA. It is perfectly structured and written. Definitely, it is very interesting to the audience! It can be published with very little editing. 

 Specific comments

Very nice and comprehensive introduction! I would suggest depicting also the concept of ecological restoration and ecosystem functioning. At least, it is possible to mention that without a holistic view on the ecosystem structure and functioning, it is not possible to undertake any successful conservation program.

Line 19: words in the keyword section should not repeat those in the title. Thus, the word “rewilding” here might be substituted for “ecological restoration.”

Lines 2-37: very nice and comprehensive story!

Lines 16: spacing before table.

Author Response

I added ecological restoration to the introduction.

I substituted ecological restoration for rewilding in the keywords.

Reviewer 4 Report

Manuscript ID: resources-2590896

Type of manuscript: Communication

Title: Rewilding the Detroit, Michigan, USA-Windsor, Ontario, Canada Metropolitan Area

Authors: John H. Hartig

Review

Aim of the paper is to overview results of the rewilding in the Detroit, Michigan, USA-Windsor, Ontario, Canada metropolitan area, based on long-term monitoring of ecosystem health indicators, and to make a summary of the benefits.

Having in mind, that rewilding of urbanized areas are of particular interest, especially after the COVID lockdowns across the globe, this manuscript is definitely warrants publication. It is clearly written and referenced. I have no negative comments, just a couple of points to consider.

First is about the article type.

According MDPI, “Communications are short articles that present groundbreaking preliminary results or significant findings that are part of a larger study over multiple years. They can also include cutting-edge methods or experiments, and the development of new technology or materials. The structure is similar to an article and there is a suggested minimum word count of 2000 words.

To my personal point of view, manuscript present intermediate version between Review and Communication. For the Review it has a surplus in references, for Communication – structure does not follow requirements.

In any case, short Discussion is required, and this section could be made from chapter 10 with adding some more references.

From the chapter 3 to 9, these could be presented as Results, to fit recommended structure of the manuscript.

Second comment is about format (as recommended is the Instructions for Authors).

·         [11] is not cited in required order, see Lines 64–66

·         Lines 93, 96, 136, 152 [ ] should be used

·         References in the Table 1, numbered from 20 to 33, are not in the order of citation: “References must be numbered in order of appearance in the text (including table captions and figure legends)”

Conflict of interest: please add if there was any role of the funding sponsors in the choice of research project; design of the study; in the collection, analyses or interpretation of data; in the writing of the manuscript; or in the decision to publish the results.

Author Response

The manuscript was submitted as a Communication per discussion with Bao Zhang of the editorial office. I was told that there was flexibility in the manuscript format for a Communication. Sections 5-8 provide evidence of rewilding and Sections 9 and 10 are a form of discussion that address remaining ecosystem challenges and benefits of rewilding and concluding thoughts. Lessons learned have been added to Section 9 to strengthen the manuscript.

Citation 11 in the literature cited has been eliminated.

The bracket "[  ]" mistakes have been fixed.

References have been reordered, including in Table 1, in a sequential fashion. I followed your guidance and included the references in Table 1 where they were first cited.  

The requested statement has been added to Conflict of Interest. 

Reviewer 5 Report

Overall I found this an interesting case study. To increase the impact of the paper beyond the two cities I have made a number of suggested modifications/additions to the text below. 

I don’t think the three paragraphs about Yellowstone are needed. Perhaps the author could note other examples of rewilding that are not urban? For example, the Scottish Highlands? Then suggest we need to more about urban

Line 48. Perhaps expand here on urban ecology. Perhaps building on the work of Dug Tallamy - https://scholar.google.com/citations?user=-vJ6qgkAAAAJ. And are there other cities that have an increased in greenspace? Eg - https://ellenmacarthurfoundation.org/circular-examples/a-biodiverse-compact-city-singapore

Line 94. Is any of this data available? Could key indicators of rewilding be shown and or discussed?

As I read on, I would either remove section 3 or merge it with section 4

Across the different taxonomic examples are there any similar patterns for success? Or lessons that could be applied beyond this case study? Likewise were there any failures?

This last point seems key to address the narrow application of this synthesis paper

Author Response

I have decreased the discussion of Yellowstone from three paragraphs to two. 

In the first paragraph, I have noted that there is considerable interest in rewilding throughout the world and added four new citations (representing Asia, Africa, Australia, and the Scottish Highlands). 

In the last paragraph of the Introduction, I have added that there is growing interest in urban rewilding and ecology, and provided three current examples - Singapore, Toronto, and Chicago. 

The State of the Strait indicators are referenced in this paper, both in peer-reviewed literature and the website where all indicator data are presented. The major ones demonstrating rewilding are the subject of the manuscript.

Relative to merging Sections 3 and 4, Section 3 documents where the trend data come from and Section 4 summarizes how environmental cleanup has catalyzed ecological recovery and rewilding. Indeed, cleanup laid the foundation for rewilding. I would prefer to leave them as separate sections.

I have added some lessons learned at the end in Section 9. This strengthens the manuscript.

Round 2

Reviewer 1 Report

Good luck!

Reviewer 5 Report

I have no future suggestions. The revisions are adaquete